# FedHyper: A Universal and Robust Learning Rate Scheduler for Federated Learning with Hypergradient Descent

**Ziyao Wang[1], Jianyu Wang[2], Ang Li[1]**
[1]University of Maryland, College Park, [2]Apple
{ziyaow,angliece}@umd.edu, {jianyuwang}@apple.com

## Abstract

The theoretical landscape of federated learning (FL) undergoes rapid evolution, but its practical application encounters a series of intricate challenges, and hyperparameter optimization is one of these critical challenges. Amongst the diverse adjustments in hyperparameters, the adaptation of the learning rate emerges as a crucial component, holding the promise of significantly enhancing the efficacy of FL systems. In response to this critical need, this paper presents FedHyper, a novel hypergradient-based learning rate scheduling algorithm for FL. FedHyper serves as a universal learning rate scheduler that can adapt both global and local learning rates as the training progresses. In addition, FedHyper not only showcases unparalleled robustness to a spectrum of initial learning rate configurations but also significantly alleviates the necessity for laborious empirical learning rate adjustments. We provide a comprehensive theoretical analysis of FedHyper's convergence rate and conduct extensive experiments on vision and language benchmark datasets. The results demonstrate that FedHyper consistently converges 1.1-3× faster than FedAvg and the competing baselines while achieving superior final accuracy. Moreover, FedHyper catalyzes a remarkable surge in accuracy, augmenting it by up to 15% compared to FedAvg under suboptimal initial learning rate settings. Our source code is released at https://github.com/ATP-1010/FedHyper.

## 1 Introduction

Federated Learning (FL) (Zhang et al., 2021; McMahan et al., 2017; Reddi et al., 2020) has emerged as a popular distributed machine learning paradigm in which a server orchestrates the collaborative training of a machine learning model across massive distributed devices. The clients only need to communicate local model updates with the server without explicitly sharing the private data. FL has been widely applied in numerous applications, such as the next-word prediction in a virtual keyboard on the smartphone (Hard et al., 2018) and hospitalization prediction for patients (Brisimi et al., 2018). However, FL presents challenges compared to conventional distributed training, including communication bottlenecks, data heterogeneity, privacy concerns, etc (Kairouz et al., 2021).

**Difficulty of Scheduling Learning Rates in FL.** In each communication round of FL, the selected clients usually perform several epochs of local stochastic gradient descent (SGD) to optimize their local models before communicating with the server in FedAvg. The server then updates the global model by the aggregated client updates Li et al. (2019); Karimireddy et al. (2020). According to this process, FL involves two types of learning rates, i.e., *global learning rate* on the server to update the global model and *local learning rate* for optimizing the local objectives on clients. They both significantly impact the model accuracy and convergence speed (Jhunjhunwala et al., 2023), while their optimal values can be influenced by various factors such as the dataset, model structure, and training stage (Reddi et al., 2020). Suboptimal learning rates can hinder model convergence, i.e., small learning rates slow down convergence while aggressive learning rates prohibit convergence (Barzilai & Borwein, 1988). Therefore, scheduling the learning rates is crucial for both speeding up convergence and improving the model performance in FL, as observed previously in centralized machine learning (You et al., 2019), Nevertheless, one of the critical challenges in FL is the data heterogene-

ity across clients Rush et al. (2023), the local optimization objectives might significantly diverge from the global one. Therefore, the server and clients need to cooperatively schedule learning rates to foster their synergy. This makes the scheduler in FL very complicated. Merely applying the optimizers in traditional machine learning, such as Adam (Kingma & Ba, 2014) and Adagrad (Lydia & Francis, 2019), to FL cannot consistently work well across different FL tasks.

In addition to the aforementioned complexities, configuring appropriate initial learning rates is also challenging for FL due to heterogeneous optimization objectives across clients. It is impractical to explore different initial learning rates due to expensive communication and computation costs. Therefore, it is necessary to design a scheduler that is robust against random initial learning rates.

**Previous Works.** Recent studies in speeding up FL have focused on adjusting the learning rate through optimizers or schedulers, primarily emphasizing the *global* model update. For instance, weight decay (Yan et al., 2022) is used to decrease the global learning rate as training progresses but exhibits much slower convergence when its initial learning rate is below the optimal value. Alternative methods, such as (Reddi et al., 2020), harness popular optimizers in centralized machine learning, i.e., Adam and Adagrad, for global updates. However, a very recent work, FEDEXP (Jhunjhunwala et al., 2023), reveals that these methods are not efficient enough in FL. FEDEXP adjusts the global learning rate based on local updates and outperforms the methods aforementioned. However, it exhibits inconsistent performance across tasks and can result in unstable accuracy in the late training stages. More importantly, these approaches mainly consider the server's perspective on global learning rate adjustments. We argue for a scheduler that integrates both global and local rates, enhancing server-client collaboration. Additionally, current solutions generally fail to ensure robustness against variations in the initial learning rate, with the exception of FEDEXP, which only maintains resilience against the initial global learning rate.

Therefore, we seek to answer a compelling research question: *How can we schedule both global and local learning rates according to the training progress and the heterogeneous local optimization objectives while ensuring robustness against the random initial learning rate?*

**Our Solution.** To tackle the question above, we design FEDHYPER, a universal and robust learning rate scheduler for FL. Specifically, FEDHYPER functions as a versatile toolbox, facilitating the scheduling of learning rates both on the server and among the clients. It encompasses (1) a *global scheduler* that schedules the global learning rate on the server; (2) a *server-side local scheduler* that tunes the local learning rate on the server; (3) a *client-side local scheduler* that adjusts the local learning rate between local epochs on the clients. These schedulers can be either independently or cooperatively applied to FL tasks, aligning with user specifications. Moreover, FEDHYPER is also seamlessly compatible with current FL optimization algorithms, thereby further improving their efficacy, such as server momentum (Liu et al., 2020) and FEDADAM (Reddi et al., 2020).

The key idea of FEDHYPER, inspired by (Chen et al., 2022; Baydin et al., 2017), is to utilize the *hypergradient* (Maclaurin et al., 2015) of the learning rate to guide the scheduling. We begin by recalling the theoretical analysis of the relationship between the optimization objective and the learning rates. The analysis reveals that the hypergradient of the learning rate is influenced by the model gradients from the current and previous training steps. Based on this insight, we formally define the hypergradients of local and global learning rates in FL by exploiting the current and historical model updates. Consequently, we design FEDHYPER, consisting of the three aforementioned schedulers. As the hypergradient of the learning rate is based on the model updates, it can precisely capture the dynamics of the training progress. Therefore, FEDHYPER can effectively schedule random initial learning rates to optimal values, making it robust against different initial learning rates. Through experiments spanning computer vision and natural language tasks, we demonstrate that FEDHYPER effectively fosters convergence, facilitates superior accuracy, and enhances the robustness against random initial learning rates. Our contributions are summarized as follows:

- We conduct a meticulous theoretical analysis for the hypergradients of local and global learning rates with regard to the gradients of local and global models.

- Based on the analysis described above, we designed FEDHYPER, a universal and robust learning rate scheduling algorithm, which can adjust both global and local learning rates and enhance the robustness against random initial learning rates.

- We propose a novel theoretical framework of FL by considering the dynamic local and global learning rates, showing that FEDHYPER is guaranteed to converge in FL scenarios.

- We conduct extensive experiments on vision and language tasks. The results demonstrate not only the efficacy of FEDHYPER to improve both convergence speed and accuracy, but also its robustness under various initial learning rate settings. FEDHYPER convergences 1.1-3x faster than FEDAVG and most competing baselines, and increases accuracy by up to 15% in various initial learning rate settings. Moreover, plugging FEDHYPER into current FL optimization methods also exhibits additional performance improvement.

## 2 RELATED WORK

**Hypergradient Descent of Learning Rates.** Hypergradient descent is commonly used to optimize the learning rate due to its integral role in the gradient descent process. Baydin et al. introduced HD (Baydin et al., 2017), an algorithm that approximates the current learning rate's hypergradient using the gradients from the last two epochs. Building on this, the Differentiable Self-Adaptive (DSA) learning rate (Chen et al., 2022) was proposed, which precomputes the next epoch's gradient and uses it, along with the current epoch's gradient, to determine the hypergradient. While DSA offers enhanced accuracy over HD, it requires greater computational resources. Moreover, studies like (Jie et al., 2022) have explored optimizing the meta-learning rate in hypergradient descent to further improve HD's efficiency.

**Hyperparameter Optimization in FL.** Hyperparameter optimization is crucial in FL. While (Wang et al., 2022) provides a benchmark, other studies like Flora (Zhou et al., 2021) and (Zhou et al., 2022) focus on hyperparameter initialization using single-shot methods. (Agrawal et al., 2021) clusters clients by data distribution to adjust hyperparameters, and FedEx (Khodak et al., 2020) employs weight sharing for streamlined optimization. Some research, such as (Kan, 2022), targets specific hyperparameters, while (Shi et al., 2022) dynamically changes the batch size during FL training. Despite these efforts, a comprehensive tool for hyperparameter optimization in FL is lacking. Our work addresses this gap, offering enhanced versatility and practicality.

## 3 PROPOSED METHOD: FEDHYPER

**Preliminaries on Hypergradient.** In general machine learning, we aim at minimizing an empirical risk function $F(\boldsymbol{w}) = \frac{1}{N}\sum_{i=1}^{N} f_i(\boldsymbol{w}; s_i)$, where $\boldsymbol{w} \in \mathbb{R}^d$ denotes the model weight and $s_i$ denotes the $i$-th sample in the training dataset. The most common way to minimize this loss function is to use gradient-based methods (Bottou, 2010), the update rule of which is given as follows:

$$\boldsymbol{w}^{(t+1)} = \boldsymbol{w}^{(t)} - \eta^{(t)}\nabla F(\boldsymbol{w}^{(t)}), \tag{1}$$

where $\eta^{(t)}$ is a time-varying scalar learning rate. The choice of $\eta$ significantly influences the training procedure. A greedy approach to set $\eta$ is to select the value that can minimize the updated loss (i.e., $\min_\eta F(\boldsymbol{w}^{(t)} - \eta\nabla F(\boldsymbol{w^{(t)}}))$). However, this is infeasible due to the need to evaluate countless possible values. Hypergradient follows this idea and allows the learning rate to be updated by a gradient as well. Specifically, when taking the derivation, one can get

$$\frac{\partial F(\boldsymbol{w}^{(t+1)})}{\partial \eta^{(t)}} = \nabla F(\boldsymbol{w}^{(t+1)}) \cdot \frac{\partial(\boldsymbol{w}^{(t)} - \eta^{(t)}\nabla F(\boldsymbol{w}^{(t)}))}{\partial \eta^{(t)}} \tag{2}$$

$$= -\nabla F(\boldsymbol{w}^{(t+1)}) \cdot \nabla F(\boldsymbol{w}^{(t)}). \tag{3}$$

Ideally, in order to get $\eta^{(t)}$, the current learning rate $\eta^{(t-1)}$ should be updated by $-\nabla F(\boldsymbol{w}^{(t+1)}) \cdot \nabla F(\boldsymbol{w}^{(t)})$. But this requires future knowledge $\nabla F(\boldsymbol{w}^{(t+1)})$. By assuming the optimal values of $\eta$ do not change much between consecutive iterations, $\eta^{(t-1)}$ is updated as follows:

$$\eta^{(t)} = \eta^{(t-1)} - \Delta_\eta^{(t)} \tag{4}$$

$$= \eta^{(t-1)} + \nabla F(\boldsymbol{w}^{(t)}) \cdot \nabla F(\boldsymbol{w}^{(t-1)}), \tag{5}$$

where $\Delta_\eta^{(t)} = -\nabla F(\boldsymbol{w}^{(t)}) \cdot \nabla F(\boldsymbol{w}^{(t-1)})$ is defined as the hypergradient. This learning rate update rule is very intuitive, allowing adaptive updates based on the training dynamics. If the inner product $\nabla F(\boldsymbol{w}^{(t)}) \cdot \nabla F(\boldsymbol{w}^{(t-1)})$ is positive, i.e., the last two steps' gradients have similar directions, then the learning rate will increase to further speedup the good progress (see Figure 1(a)). If the inner product is negative, i.e., two consecutive gradients point to drastically diverged directions, then the learning rate will decrease to mitigate oscillations (see Figure 1(b)).

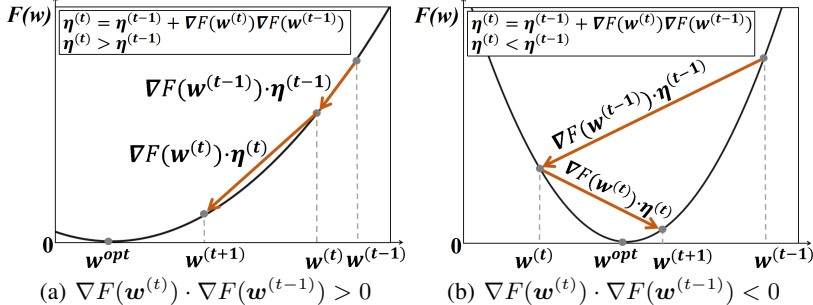

$$\text{(a) } \nabla F(\boldsymbol{w}^{(t)}) \cdot \nabla F(\boldsymbol{w}^{(t-1)}) > 0 \qquad \text{(b) } \nabla F(\boldsymbol{w}^{(t)}) \cdot \nabla F(\boldsymbol{w}^{(t-1)}) < 0$$

Figure 1: Two situations in hypergradient descent. Hypergradient descent adjusts the learning rate according to the inner product of gradients in the last two iterations.

**Applying Hypergradient to FL.** In FL, the goal is to optimize a global objective function $F(\boldsymbol{w})$ which is defined as a weighted average over a large set of local objectives:

$$F(\boldsymbol{w}) = \frac{1}{M} \sum_{m=1}^{M} F_m(\boldsymbol{w}), \tag{6}$$

where $M$ is the number of clients and $F_m(\boldsymbol{w}) = \frac{1}{N_m} \sum_n f_m(\boldsymbol{w}; s_n)$ is the local loss function of the $m$-th client. Due to privacy concerns, the server cannot access any client data and clients can neither share data with others. FEDAVG (McMahan et al., 2017) is the most popular algorithm to solve this problem. It allows clients to perform local training on their own local dataset. A central coordinating server only aggregates the model weight changes to update the server model:

$$\boldsymbol{w}^{(t+1)} = \boldsymbol{w}^{(t)} - \alpha^{(t)} \frac{1}{M} \sum_{m=1}^{M} \Delta_m^{(t)}, \tag{7}$$

where $\alpha^{(t)}$ is defined as the server learning rate, and $\Delta_m^{(t)}$ is the local model changes. Moreover, the local update rule on each client is:

$$\boldsymbol{w}_m^{(t,k+1)} = \boldsymbol{w}_m^{(t,k)} - \beta^{(t,k)} g_m(\boldsymbol{w}_m^{(t,k)}), \tag{8}$$

$$\Delta_m^{(t)} = \boldsymbol{w}_m^{(t,0)} - \boldsymbol{w}_m^{(t,\tau)} = \boldsymbol{w}^{(t)} - \boldsymbol{w}_m^{(t,\tau)}, \tag{9}$$

where $\beta^{(t,k)}$ is the local learning rate, $\boldsymbol{w}_m^{(t,k)}$ denote the local model at $m$-th client at the $t$-th global round and $k$-th local step, and $\boldsymbol{w}_m^{(t,0)} = \boldsymbol{w}^{(t)}$ is the initial model at each round.

Comparing the update rules of FEDAVG with vanilla gradient descent, the learning rate scheduling for FedAvg can be way more complicated. When applying the vanilla hypergradient idea on the server learning rate $\alpha$, one will find that the global gradient $\nabla F(\boldsymbol{w})$ is not available at all; when applying the idea on the client learning rate, the derivative with respect to $\beta^{(t,k)}$ is nearly intractable.

In order to address these challenges, we develop FEDHYPER, a learning rate scheduler based on hypergradient descent specifically designed for FL. FEDHYPER comprises a global scheduler FEDHYPER-G, a server-side local scheduler FEDHYPER-SL, and a client-side local scheduler FEDHYPER-CL. The general learning rates update rules are given below:

$$\text{FEDHYPER-G: } \alpha^{(t)} = \alpha^{(t-1)} - \Delta_\alpha^{(t-1)}, \tag{10}$$

$$\text{FEDHYPER-SL: } \beta^{(t,0)} = \beta^{(t-1,0)} - \Delta_{\beta,s}^{(t-1)}, \tag{11}$$

$$\text{FEDHYPER-CL: } \beta^{(t,k)} = \beta^{(t,k-1)} - \Delta_{\beta,c}^{(t,k-1)}. \tag{12}$$

In the following subsections, we will specify the exact expressions of each hypergradient.

### 3.1 FEDHYPER-G: USING HYPERGRADIENT FOR THE GLOBAL LEARNING RATE

We first provide FEDHYPER-G to adjust the global learning rate $\alpha$ based on the hypergradient descent. According to Eq. (1), the global learning rate for the $t$th round $\alpha^{(t)}$ should be computed by

the last global learning rate $\alpha^{(t-1)}$ and global gradients in the two preceding rounds. Nevertheless, in accordance with the global model updating strategy in Eq. (7), the global model is updated by the aggregation of local model updates rather than gradients. Therefore, in FEDHYPER-G, we treat the aggregation of the local model updates as the pseudo global gradient $\Delta^{(t)}$:

$$\Delta^{(t)} = \frac{1}{M} \sum_{m=1}^{M} \Delta_m^{(t)} \tag{13}$$

Then, we replace the gradients in Eq. (5) with global model updates and obtain the $\Delta_\alpha^{(t-1)}$ in Eq. (10) for the global scheduler as follows:

$$\Delta_\alpha^{(t-1)} = -\Delta^{(t)} \cdot \Delta^{(t-1)}, \tag{14}$$

$$\alpha^{(t)} = \alpha^{(t-1)} + \Delta^{(t)} \cdot \Delta^{(t-1)} \tag{15}$$

In addition, to ensure convergence and prevent gradient exploration, we clip $\alpha^{(t)}$ by a preset bound value $\gamma_\alpha$ in each round:

$$\alpha^{(t)} = \min\{\max\{\alpha^{(t)}, \frac{1}{\gamma_\alpha}\}, \gamma_\alpha\} \tag{16}$$

Through experiments, we have found that $\gamma_\alpha = 3$ yields consistent and stable performance across various tasks and configurations. Therefore, we generally do not need to further adjust $\gamma_\alpha$.

### 3.2 FEDHYPER-SL & FEDHYPER-CL: USING HYPERGRADIENT FOR THE CLIENT LEARNING RATE

**FedHyper-SL.** The server-side local scheduler adopts the same hypergradient as the global scheduler, i.e., $\Delta_{\beta,s}^{(t-1)} = \Delta_\alpha^{(t-1)}$. Thus, the local learning rates are updated by the server as follows:

$$\beta^{(t,0)} = \beta^{(t-1,0)} + \Delta^{(t)} \cdot \Delta^{(t-1)}. \tag{17}$$

It adjusts all the selected local learning rates from the server synchronously. These local learning rates are also clipped by a preset bound $\gamma_\beta$:

$$\beta^{(t,0)} = \min\{\max\{\beta^{(t,0)}, \frac{1}{\gamma_\beta}\}, \gamma_\beta\}. \tag{18}$$

The updated local learning rates $\beta^{(t,0)}$ are sent to clients *after* $t$-th round and are used by clients in the $t+1$-th round. Like FEDHYPER-G, we have experimentally set $\gamma_\beta$ to 10. This bound will also be used for $\beta^{(t,k)}$ in FEDHYPER-CL.

**FedHyper-CL.** Adjusting the local learning rate from the clients is a more fine-grained scheduling strategy since it adjusts learning rates for each local batch, while the server only performs such adjustments between communication rounds. This client-centric approach not only enhances convergence speed but also outperforms both FEDHYPER-G and FEDHYPER-SL in terms of accuracy. According to the hypergradient descent in Eq. (5) and local model update rule Eq. (8), the local learning rate for the $k$-th epoch on client $m$ can be updated as follows:

$$\Delta_{\beta,c}^{(t,k-1)} = -g_m(\boldsymbol{w}_m^{(t,k)}) \cdot g_m(\boldsymbol{w}_m^{(t,k-1)}). \tag{19}$$

Nevertheless, directly applying Eq. (19) to local learning rates can lead to an imbalance in learning rates across clients, which may lead to convergence difficulty. To address this issue, and draw inspiration from our global scheduler, we incorporate global model updates to regulate the local learning rate optimization. Specifically, we augment $\Delta_{\beta,c}^{(t,k-1)}$ by adding the component $\frac{1}{K} \cdot g_m(\boldsymbol{w}_m^{t,k}) \cdot \Delta^{t-1}$, where $\Delta^{t-1}$ is the global model update in the last round and $K$ is the number of local SGD steps:

$$\beta^{(t,k)} = \beta^{(t,k-1)} + g_m(\boldsymbol{w}_m^{(t,k)}) \cdot g_m(\boldsymbol{w}_m^{(t,k-1)}) + \frac{1}{K} \cdot g_m(\boldsymbol{w}_m^{(t,k)}) \cdot \Delta^{(t-1)}. \tag{20}$$

The term $g_m(\boldsymbol{w}_m^{(t,k)}) \cdot \Delta^{(t-1)}$ will be positive if $g_m(\boldsymbol{w}_m^{(t,k)})$ and $\Delta^{t-1}$ have similar direction, otherwise it is negative. The coefficient $\frac{1}{K}$ acts to normalize the magnitude of $g_m(\boldsymbol{w}_m^{(t,k)}) \cdot \Delta^{(t-1)}$ such

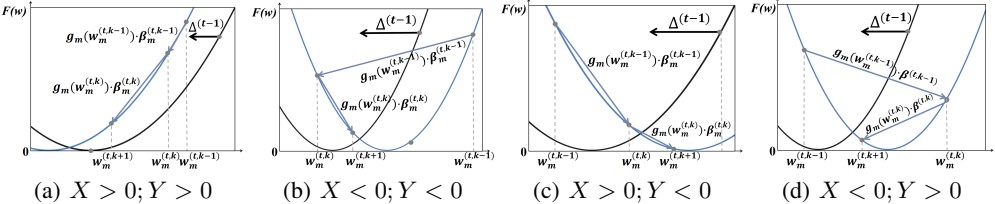

Figure 2: Four situations in client-side local learning rate scheduling.

that it aligns with the scale of $g_m(\boldsymbol{w}_m^{(t,k)}) \cdot g_m(\boldsymbol{w}_m^{(t,k-1)})$. This normalization is essential given that while $g_m(\boldsymbol{w}_m^{(t,k)})$ is the gradient of a single SGD step, $\Delta^{t-1}$ is the average over $K$ local SGD steps. By integrating this term, we increase $\beta^{(t,k)}$ when the local model update is directionally consistent with the global model update; conversely, we decrease it when they exhibit significant discrepancies. It is pivotal to prevent certain clients from adopting excessively high learning rates that could potentially impact convergence.

Here we define $X = g_m(\boldsymbol{w}_m^{(t,k)}) \cdot g_m(\boldsymbol{w}_m^{(t,k-1)}); Y = \frac{1}{K} \cdot g_m(\boldsymbol{w}_m^{(t,k)}) \cdot \Delta^{(t-1)}$. The local learning rate in the following four situations will be adjusted as: (1) $X > 0; Y > 0$ as in Figure 2(a), $\beta^{(t,k)}$ will increase; (2) $X < 0; Y < 0$ as in Figure 2(b), $\beta^{(t,k)}$ will decrease; (3) $X > 0; Y < 0$ as in Figure 2(c), $\beta^{(t,k)}$ will increase if $|X| > |Y|$; otherwise, it will decrease; (4) $X < 0; Y > 0$ as in Figure 2(d), $\beta^{(t,k)}$ will increase if $|X| < |Y|$; otherwise, it will undergo a decrease.

The client-side local scheduler integrates insights from both global and local training dynamics. By leveraging global model updates, it steers the local learning rate optimization effectively. The full algorithm of FEDHYPER is obtained in Appendix C.

## 4    CONVERGENCE GUARANTEE

In this section, we analyze the convergence of FEDHYPER considering dynamic global and local learning rates. Our analysis demonstrate that FEDHYPER can ensure convergence given a sufficient number of training epochs under some standard assumptions (Wang et al., 2020b).

According to the unified convergence analysis in FEDNOVA (Wang et al., 2020b), the convergence of an FL framework based on the objective in Eq. (6) can be proved with some basic assumptions, which we list in Appendix A.

In addition, we have the following upper and lower bounds for global and local learning rates:

**Constraint 1** The learning rates satisfiy $\frac{1}{\gamma_\alpha} \leq \alpha^{(t)} \leq \gamma_\alpha$ and $\frac{1}{\gamma_\beta} \leq \beta^{(t,k)} \leq \gamma_\beta$, where $\gamma_\alpha, \gamma_\beta > 1$.

**Theorem 1** Under Assumption 1 to 3 in Appendix A and constraint 1 above, we deduce the optimization error bound of FEDHYPER from the convergence theorem of FEDNOVA.

$$\min_{t \in [T]} \mathbb{E}\|\nabla F(\boldsymbol{w}^{(t)})\|^2 < O\left(\frac{P}{\sqrt{MkT}} + \frac{Q}{kT}\right), \tag{21}$$

where quantities $P, Q$ are defined as follows:

$$P = \left(\sum_{m=1}^{M} \frac{\gamma_\beta^4 \sigma^2}{M} + 1\right)\gamma_\alpha, \tag{22}$$

$$Q = \sum_{m=1}^{M} \left[[(k-1)\gamma_\beta^4 + 1]^2 - \frac{1}{\gamma_\beta^2}\right]\sigma^2 + M\rho^2\left[[(k-1)\gamma_\beta^2 + 1]^2 - \frac{1}{\gamma_\beta^2}[\frac{k-1}{\gamma_\beta^2} + 1]\right]. \tag{23}$$

Where $\sigma$ and $\rho$ are constants that satisfy $\sigma^2, \rho^2 \geq 0$. The bound defined by Eq. (21), (22), and (23) indicates that when the number of rounds $T$ tends towards infinity, the expectation of $\nabla F(\boldsymbol{w}^{(t)})$ tends towards 0. We further provide a comprehensive proof of Theorem 1 in Appendix A.

## 5 EXPERIMENTS

We evaluate the performance of FEDHYPER on three benchmark datasets, including FMNIST (Xiao et al., 2017) and CIFAR10 (Krizhevsky et al., 2009) for image classification, and Shakespeare (McMahan et al., 2017) for the next-word prediction task. To evaluate the efficacy of adapting the global learning rate, we compare FEDHYPER-G against four baselines: FEDADAGRAD (Reddi et al., 2020), FEDADAM (Reddi et al., 2020), FEDEXP (Jhunjhunwala et al., 2023) and global learning rate decay with a 0.995 decay factor. We employ the across-round local learning rate step decay with also a 0.995 decay factor as the baseline to compare with the server-side local scheduler FEDHYPER-SL. As for client-side local scheduler FEDHYPER-CL, we opt for FEDAVG paired with local SGD and local Adam as the comparative baselines.

**Experimental Setup.** We partition FMNIST and CIFAR10 dataset into 100 clients following a Dirichlet distribution with $\alpha = 0.5$ as presented in(Wang et al., 2020a). We directly partition the inherently non-iid Shakespeare dataset to 100 clients as well. In each communication round, the server randomly selects 10 clients to perform local SGD and update the global model. For FMNIST, we utilize a CNN model, while for CIFAR10, we deploy a ResNet-18 (He et al., 2016). For the next-word prediction task on Shakespeare, we implement a bidirectional LSTM model (Graves et al., 2005). We set our global bound parameter $\gamma_\alpha$ at 3 and the local bound $\gamma_\beta$ at 10 in all experiments. In order to achieve global model convergence, we conduct 50, 200, and 600 communication rounds for FMNIST, CIFAR10, and Shakespeare, respectively.

### 5.1 FEDHYPER COMPREHENSIVELY OUTPERFORMS FEDAVG AND BASELINES

**Performance of FedHyper-G.** As Figure 3(a) illustrates, the results underscore the superiority of FEDHYPER-G over the current global optimizers and schedulers in FL. Across all three datasets, FEDHYPER-G achieves faster convergence rates in the early training stage (e.g., rounds 0 to 25 in FMNIST, rounds 0 to 80 in CIFAR10, and rounds 0 to 300 in Shakespeare) than all baselines. Moreover, FEDHYPER-G also maintains an equal or superior accuracy than baselines.

Such results corroborate our intuition in Section 3: FEDHYPER can expedite the training process when the model remains far from the optimum and enhance the accuracy. Additionally, FEDHYPER-G increases accuracy by 0.45% 0.58%, and 0.66% compared to the best baseline in FMNIST, CIFAR10, and Shakespeare, respectively. This underscores the consistent performance improvement of FEDHYPER-G across varied tasks. Although FEDEXP shows comparable performance as FEDHYPER-G in FMNIST and CIFAR10 for image classification, FEDHYPER-G outperforms FEDEXP on Shakespeare dataset.

**Performance of FedHyper-SL.** We compare FEDHYPER-SL with FEDAVG and across-round local learning rate decay, i.e., Decay-SL. As Figure 3(b) shows, FEDHYPER-SL can potentially lead to comparable, if not superior, convergence speed and accuracy when contrasted with FEDHYPER-G. This observation is particularly pronounced in the CIFAR10 and Shakespeare datasets. For instance, in the Shakespeare dataset, FEDHYPER-SL achieves a 50% test accuracy approximately by the 150th round, whereas FEDHYPER-G reaches a similar accuracy near the 250th round. A plausible explanation for this might be that the local learning rate impacts every local SGD iteration, in contrast to the global scheduler, which only impacts the aggregation process once per round.

**Performance of FedHyper-CL.** Figure 3(c) exhibits the efficacy of the client-side local scheduler. Not only does FEDHYPER-CL outperform the baselines, but it also manifests superior performance compared to our other two schedulers, as analyzed in Section 3. Specifically, in FMNIST and CIFAR10, it outperforms FEDAVG with SGD and Adam, with improvements of 0.44% and 1.18%, respectively. Regarding the Shakespeare dataset, FEDHYPER-CL excels over FEDAVG with local SGD by 1.33%, though its performance aligns closely with that of FEDAVG with local Adam. Overall, FEDHYPER-CL consistently demonstrates commendable results across diverse tasks.

Additionally, FEDHYPER-CL also shows a better performance compared to FEDHYPER-G and FEDHYPER-SL, most notably in terms of convergence speed. For example, FEDHYPER-CL achieves 52.08% accuracy, converging approximately by the 100th round. In contrast, FEDHYPER-G and FEDHYPER-SL yield accuracies of 51.08% and 51.70% respectively, with a protracted convergence around the 300th round. It is noteworthy that FEDHYPER-CL does incur a higher computa-

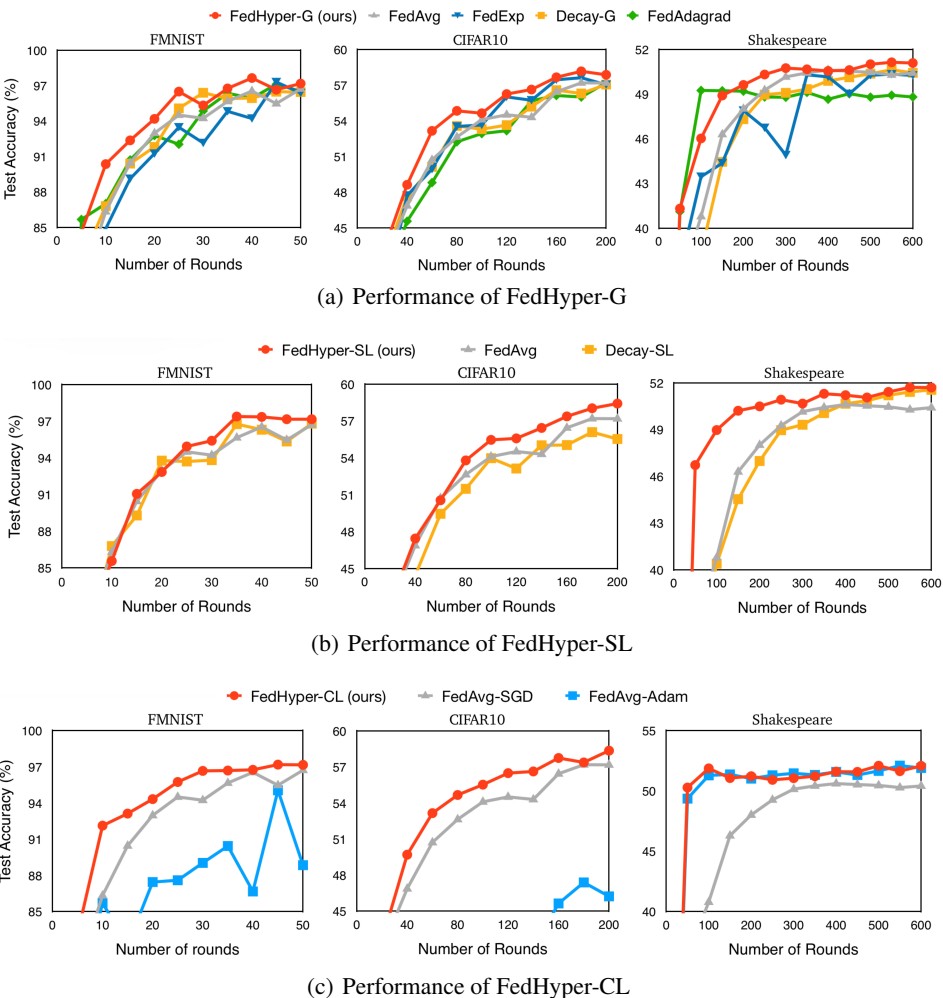

Figure 3: Comparison of FedHyper with baselines. FEDHYPER consistently gives faster convergence compared to baselines with superior performance.

tional overhead compared to the other two methods in FEDHYPER. Consequently, a comprehensive discussion regarding the selection of three schedulers will be presented in the Appendix B.

## 5.2 DISCUSSION AND ABLATIONS OF FEDHYPER

**Robustness of FEDHYPER Against Initial Learning Rates.** As discussed in Section 1, identifying the optimal initial learning rates in FL presents a significant challenge. FEDHYPER facilitates faster scheduling of the learning rates, showing its versatility to attain desired accuracy even in scenarios characterized by suboptimal initial learning rates, especially when contrasted with FEDAVG.

We first conduct experiments on CIFAR10 and Shakespeare to evaluate the robustness of FEDHYPER against varied initial learning rates. Specifically, we employ five distinct initial global learning rates: 0.5, 0.75, 1, 1.5, and 2. Additionally, we also consider five separate initial local learning rates: 0.001, 0.005, 0.01, 0.05, and 0.1. We first train CIFAR10 and Shakespeare using FEDAVG by applying the aforementioned learning rates. The final accuracy is depicted in Figure 4(a) and Figure 4(c). The darker red shades represent higher final test accuracy.

Following the initial experiments, we apply FEDHYPER in conjunction with both the global scheduler and the client-side local scheduler, denoted as FEDHYPER-G + FEDHYPER-CL, employing the identical initial learning rates as mentioned previously. The results are illustrated in Figure 4(b) and Figure 4(d), the shades of green serve as indicators of the magnitude of accuracy augmentation, with the darker green hues representing more substantial improvements in accuracy.

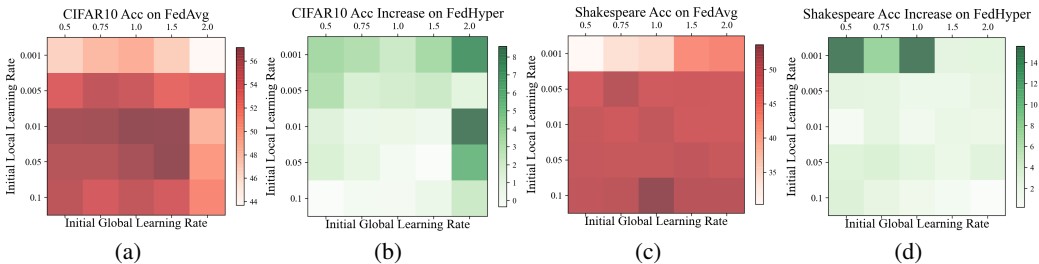

Figure 4: FEDHYPER's robustness against diverse initial learning rates.

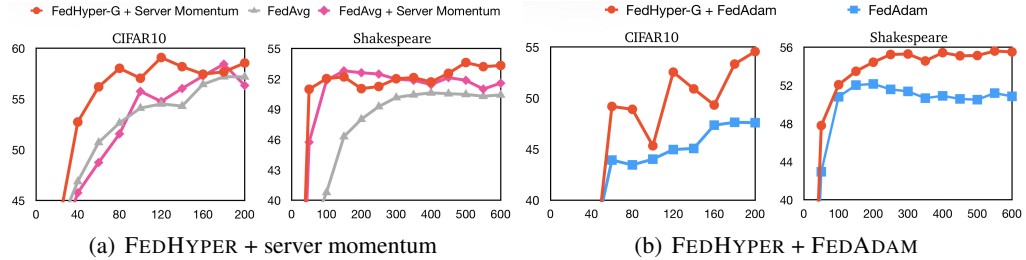

Figure 5: Integration of FEDHYPER with server momentum and FEDADAM.

The results in Figure 4 show that FEDHYPER improves accuracy by 1% - 5% over FEDAVG for most initial learning rate configurations. Notably, FEDHYPER is particularly effective at enhancing the accuracy of FEDAVG when it is faced with suboptimal initial learning rates. For instance, in the Shakespeare dataset (see Figure 4(c)), FEDAVG excels with higher initial local learning rates ($> 0.005$), but underperforms at suboptimal initial local learning rates like $0.001$. In such scenarios, Figure 4(d) demonstrates FEDHYPER's ability to substantially boost accuracy, notably in cases with a 0.001 initial local learning rate. When paired with a 0.5 initial global rate, FEDAVG achieves 30% accuracy, whereas FEDHYPER raises it by almost 15% to 45.77%. The robustness of FEDHYPER against random initial learning rates can be visually demonstrated in Figure 4, i.e., areas characterized by a lighter shade of red (i.e., indication of lower accuracy) in Figure 4(a) and Figure 4(c) transform into darker shades of green (representation of significant accuracy augmentation) in Figure 4(b) and Figure 4(d). The robustness manifested by FEDHYPER thus mitigates the necessity for meticulous explorations to initial learning rates within the FL paradigm.

**Integrating FedHyper with Existing Optimizers.** We evaluate the efficacy of FEDHYPER-G when it is integrated with existing global optimization algorithms, i.e., server momentum (Liu et al., 2020) and FEDADAM (Reddi et al., 2020). The results in Figure 5 underscore that FEDHYPER is capable of improving the performance of these widely-adopted FL optimizers. For instance, FEDHYPER-G together with server momentum shows superior performance over the combination of FEDAVG and server momentum. Moreover, as Figure 5(b) depicts, when FEDHYPER-G is paired with FEDADAM for the CIFAR10, there is a marked enhancement in performance. These findings demonstrate that FEDHYPER possesses the versatility to serve not merely as a universal scheduler, but also as a universally applicable enhancement plugin that can elevate the capabilities of existing optimizers.

## 6 CONCLUSION

In this paper, we introduced FEDHYPER, a universal and robust learning rate scheduling algorithm rooted in hypergradient descent. FEDHYPER can optimize both global and local learning rates throughout the training. Our experimental results have shown that FEDHYPER consistently outperforms baseline algorithms in terms of convergence rate and test accuracy. Our ablation studies demonstrate the robustness of FEDHYPER under varied configurations of initial learning rates. Furthermore, our empirical evaluations reveal that FEDHYPER can seamlessly integrate with and augment the performance of existing optimization algorithms.

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

## A  PROOF OF THEOREM 1: CONVERGENCE OF FEDHYPER

According to theoretical analysis of FEDNOVA (Wang et al., 2020b), an FL framework that follows the update rule in Eq. (13) will converge to a stationary point, and the optimization error will be bounded with the following assumptions:

**Assumption 1** (Smoothness). Each local objective function is Lipschitz smooth, that is, $\|\nabla F_m(x) - \nabla F_m(y)\| \leq L\|x - y\|, \forall m \in \{1, 2, ..., M\}$.

**Assumption 2** (Unbiased Gradient and Bounded Variance). The stochastic gradient at each client is an unbiased estimator of the local gradient: $E_\xi[g_i(x|\xi)] = \nabla F_m(x)$ and has bounded variance $E_\xi[\|g_m(x|\xi) - \nabla F_m(x)\|^2] \leq \sigma^2, \forall m \in \{1, 2, ..., M\}, \sigma^2 \geq 0$.

**Assumption 3** (Bounded Dissimilarity). Existing constants $\psi^2 \geq 1, \rho^2 \geq 0$ such that $\Sigma_{m=1}^M \frac{1}{M}\|\nabla F_m(x)\|^2 \leq \psi^2\|\Sigma_{m=1}^M \frac{1}{M}\nabla F_m(x)\|^2$. If local functions are identical to each other, then we have $\psi^2 = 1, \rho^2 = 0$.

where we adhere to our setting that each client contributes to the global model with the equal weight $\frac{1}{M}$. Then we can rewrite the optimization error bound as follows:

$$\min_{t \in [T]} \mathbb{E}\|\nabla F(w^{(t)})\|^2 \leq O\left(\frac{\alpha^{(t)}}{\sqrt{MkT}}\right) + O\left(\frac{A\sigma^2}{\sqrt{MkT}}\right) + O\left(\frac{MB\sigma^2}{kT}\right) + O\left(\frac{MC\rho^2}{kT}\right), \quad (24)$$

Where A, B, and C are defined by:

$$A = \alpha^{(t)} \sum_{m=1}^M \frac{\|a_m\|_2^2}{M\|a_m\|_1^2}, B = \sum_{m=1}^M \frac{1}{M}(\|a_m\|_2^2 - a_{m,-1}^2), C = \max_m\{\|a_m\|_1^2 - \|a_m\|_1 a_{m,-1}\}. \quad (25)$$

where $a_m$ is a vector that can measure the local model update during local SGD, where the number of $k$-th value of $a_m$ is $a_m[k] = \frac{\beta^{(t,k)}}{\beta^{(t,0)}}$.

Note that we have Bound 1 on global learning rate that $\alpha^{(t)} = \min\{\max\{\alpha^t, \frac{1}{\gamma_\alpha}\}, \gamma_\alpha\}$, so we have the upper and lower bound for $\alpha^{(t)}$ as follows:

$$\frac{1}{\gamma_\alpha} \leq \alpha^{(t)} \leq \gamma_\alpha, \quad (26)$$

For the local learning rate, we have $\beta^{(t,k)} = \min\{\max\{\beta^{(t,k)}, \frac{1}{\gamma_\beta}\}, \gamma_\beta\}$. Therefore, the maximum value of ratio $\frac{\beta^{(t,k)}}{\beta^{(t,0)}}$ is $\gamma_\beta^2$, when $\beta^{(t,k)} = \gamma_\beta$, and $\beta^{(t,0)} = \frac{1}{\gamma_\beta}$. Accordingly, the minimum of $\frac{\beta^{(t,k)}}{\beta^{(t,0)}}$ is $\frac{1}{\gamma_\beta^2}$. We can derive the upper and lower bound also for $\|a_m\|_1$ and $\|a_m\|_2$ as follows:

$$\frac{1}{\gamma_\beta^2} \leq a_{m,k} \leq \gamma_\beta^2,$$

$$\frac{k-1}{\gamma_\beta^2} + 1 \leq \|a_m\|_1 \leq (k-1)\gamma_\beta^2 + 1,$$

$$\frac{k-1}{\gamma_\beta^4} + 1 \leq \|a_m\|_2 \leq (k-1)\gamma_\beta^4 + 1, \quad (27)$$

$$\frac{\|a_m\|_2}{\|a_m\|_1} \leq \gamma_\beta^2,$$

Then, we apply Eq. (26) and (27) to the first item of Eq. (25), and get:

$$O\left(\frac{\alpha^{(t)}}{\sqrt{MkT}}\right) \leq O\left(\frac{\gamma_\alpha}{\sqrt{MkT}}\right), \quad (28)$$

Then, we apply Eq. (26) and (27) to Eq. (25) and redefine $A$ $B$ and $C$:

$$
\begin{aligned}
A &= \alpha^t \sum_{m=1}^{M} \frac{\|a_m\|_2^2}{M \|a_m\|_1^2} \\
&\leq \gamma_\alpha \sum_{m=1}^{M} \frac{\|a_m\|_2^2}{M \|a_m\|_1^2} \\
&\leq \gamma_\alpha \sum_{m=1}^{M} \frac{\gamma_\beta^4}{M},
\end{aligned}
\tag{29}
$$

$$
\begin{aligned}
B &= \sum_{m=1}^{M} \frac{1}{M}(\|a_m\|_2^2 - a_{m,-1}^2) \\
&< \sum_{m=1}^{M} \frac{1}{M}[[(k-1)\gamma_\beta^4 + 1]^2 - \frac{1}{\gamma_\beta^2}],
\end{aligned}
\tag{30}
$$

$$
\begin{aligned}
C &= \max_m \{\|a_m\|_1^2 - \|a_m\|_1 a_{m,-1}\} \\
&< [(k-1)\gamma_\beta^2 + 1]^2 - \frac{1}{\gamma_\beta^2}[\frac{k-1}{\gamma_\beta^2} + 1],
\end{aligned}
\tag{31}
$$

Then, we can combine the first and second items of Eq. (24) and get the new bound:

$$
\min_{t \in [T]} \mathbb{E}\|\nabla F(W^t)\|^2 \leq O\left(\frac{P}{\sqrt{MkT}} + \frac{Q}{kT}\right),
\tag{32}
$$

where $P$ is defined by:

$$
\begin{aligned}
P &= \gamma_\alpha + A\sigma^2 \\
&= (\sum_{m=1}^{M} \frac{\gamma_\beta^4 \sigma^2}{M} + 1)\gamma_\alpha,
\end{aligned}
\tag{33}
$$

and Q is defined by:

$$
\begin{aligned}
Q &= MB\sigma^2 + MC\rho^2 \\
&= \sum_{m=1}^{M}[[(k-1)\gamma_\beta^4 + 1]^2 - \frac{1}{\gamma_\beta^2}]\sigma^2 + M\rho^2[[(k-1)\gamma_\beta^2 + 1]^2 - \frac{1}{\gamma_\beta^2}[\frac{k-1}{\gamma_\beta^2} + 1]].
\end{aligned}
\tag{34}
$$

Note that we use the upper bound of $A, B, C$ here. Now we have completed the proof of Theorem 1.

## B SUPPLEMENTARY EXPERIMENTAL RESULTS

### B.1 LEARNING RATE CURVE OF FEDHYPER

As we analyzed in Section 3, FEDHYPER adjusts the learning rates in a way that the learning rates increase in the former training stages and decrease in the latter stages. It also aligns with our analysis of the relationship between the convergence and the value of learning rates in Figure 1. To illustrate this point, we visualize the change in global learning rate through 0-100 epochs in Figure 6. The global learning rate of FEDHYPER-G increases from round 0 to 15, and starts to decrease. The value is greater than 1 in the first 50 epochs and less than 1 in the following 50 epochs. As for FEDEXP, the global learning rate value fluctuates between 1.0 and 1.5.

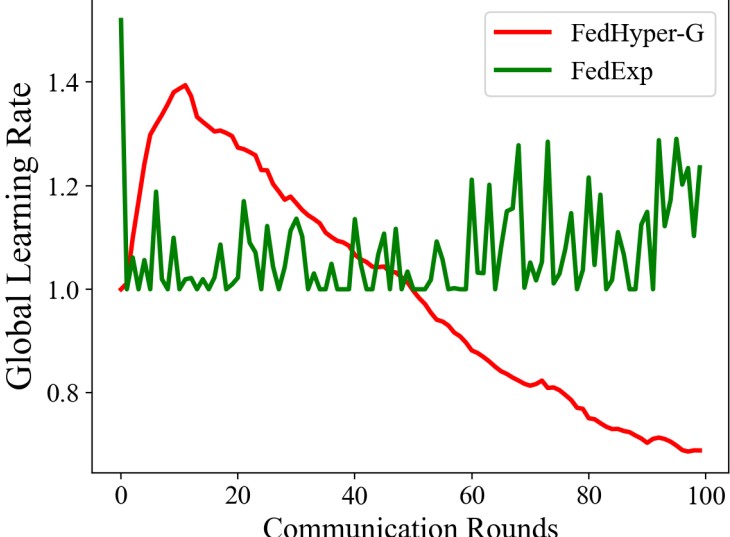

Figure 6: Comparison of global learning rate curve between FEDHYPER and FEDEXP.

## B.2 THE TIME OVERHEAD OF FEDHYPER

FEDHYPER involves utilizing the inner product of gradients to modify the learning rate. It may raise concern that FEDHYPER incurs additional computational expenses compared to the more basic FedAvg. Here we claim that hypergradient computing in FEDHYPER only involves a small amount of time overhead, which is trivial compared to the original time consumption of FEDAVG. To prove this, we have included an analysis of the time cost of our three schedulers and baselines. We ran the experiments on time cost of our three schedulers and baselines. The following results in Table 1 are the time cost (in seconds) of different optimization algorithms when training CIFAR10 for 200 communication rounds:

| FedHyper-G | FedHyper-CL | FedAvg | FedExp | FedAdam | FedAdagrad |
|---|---|---|---|---|---|
| 44060 | 45780 | 44031 | 45193 | 44045 | 43917 |

Table 1: Total time consumption in second of FEDHYPER and baselines

Different global schedulers have similar running times, however, the schedulers in FEDHYPER can reach convergence faster than baselines, so we can have less cost than our baselines ultimately. Specifically, FEDHYPER-CL only increases less than 5% computation cost compared with FE-DAVG(SGD) but gets convergence up to 3 times faster than SGD. Therefore, FEDHYPER's faster convergence offsets this slight increase in computation time.

## B.3 THE IMPACT OF NON-IID LEVEL

The data distribution on clients also affects FL performance. In Table 2, we display the final accuracy of global models with FEDAVG and FEDHYPER in iid and non-iid data, and also different $\alpha$ in non-iid Dirichlet distribution. We can conclude from the table that FEDHYPER contributes more to non-iid settings, especially with relatively small $\alpha$ numbers. The accuracy increase of $\alpha = 0.25$ is 0.80% in FMNIST and 1.55% in CIFAR10, which is the highest among the three $\alpha$ values. This might be attributed to the client-side local scheduler we designed that adopts the global updates to restrict the increasing of local learning rates on some clients that might hinder the global convergence because of the data heterogi.

| | FMNIST | | | | CIFAR10 | | | |
|---|---|---|---|---|---|---|---|---|
| | IID | 0.75 | 0.5 | 0.25 | IID | 0.75 | 0.5 | 0.25 |
| FedAvg | 98.62% | 96.35% | 96.36% | 97.00% | 67.14% | 56.71% | 57.14% | 53.55% |
| FedHyper | 98.92% | 97.03% | 96.97% | 97.80% | 67.45% | 57.18% | 58.42% | 55.10% |

Table 2: Accuracy on FedHyper in different $\alpha$ values

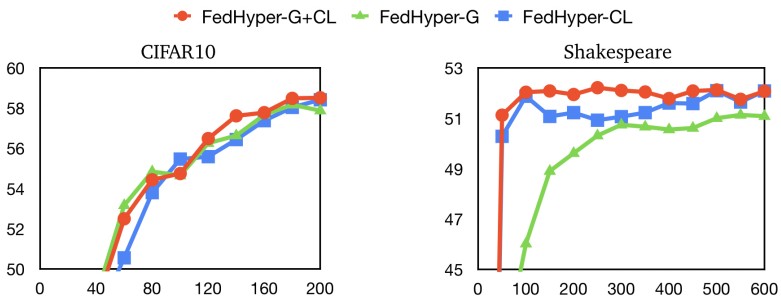

Figure 7: Cooperation of FEDHYPER-G and FEDHYPER-CL.

## B.4 COMBINATION OF FEDHYPER-G AND FEDHYPER-CL

We show the results of FEDHYPER-G, FEDHYPER-SL, and FEDHYPER-CL work alone in Figure 3 and show that they can both outperform baselines that optimize the global or local training from the same dimension (e.g. both work on the server). However, FEDHYPER has another advantage over baselines, that is, the ability to adjust both global and local learning rates in one training process. To support this, we run experiments on CIFAR10 and Shakespeare by applying both FEDHYPER-G and FEDHYPER-CL, called FEDHYPER-G+CL. We display the results in Figure 7 and compare it with FEDHYPER-G and FEDHYPER-CL. The results show that FEDHYPER-G+CL is still able to outperform both of the single adjusting algorithms, which indicates that the performance of FED-HYPER algorithms can superpose each other. This makes FEDHYPER more flexible and can be suitable for different user needs. Here we do not show the results of combining FEDHYPER-G and FEDHYPER-SL because they use the same hypergradient to adjust different learning rates. So the superposition effect is not obvious.

## B.5 HOW TO USE FEDHYPER IN REAL FL PROJECTS

We have three schedulers in FEDHYPER framework. However, not all of them are needed in real FL projects. We highly recommend FL trainers select suitable algorithms according to their needs and budgets. Here we provide some suggestions on the algorithm selection in specific scenarios.

**FedHyper-G only** when the trainer has a tight budget of computational resources on clients, e.g., when performing FL on edge devices, mobile terminal devices, or low-memory GPUs.

**FedHyper-SL only** has a similar scenario with FEDHYPER-G only. One thing difference is that it adds some extra communication costs in sending the local learning rates. Therefore, if the trainer does not have a bottleneck in communication cost, she can choose freely between FEDHYPER-G only and FEDHYPER-G only while considering the features of the specific task (i.e., more sensitive to global or local learning rates).

**FedHyper-CL only** when the trainer has a tight budget of computational resources on the server but a loose budget on clients, e.g., FL service providers.

**FedHyper-G and FedHyper-CL** when the trainer has loose budgets of computational resources on both server and clients, e.g. distributed large model training.

---

**Algorithm 1** Workflow of FEDHYPER

---

**Input:** Initial Global Model $W^0$, Number of Communication Rounds $T$, Number of Selected Clients each Round $M$, Initial Global Learning Rate $\alpha^0$, Initial Local Learning Rate each Round $\beta^0 = \beta^1 = \beta^0 = ... = \beta^{T-1}$, Local Epoch number $K$, Local batches $\xi$;

**Output:** Trained Global Model $W^T$;

 1: **for** $t$ in $0, 1, ..., T-1$ **do**
 2:     Server send $W^t$ and $\beta^t$ to all selected clients.

**Clients: FedHyper-CL**

 3:     Compute global model update $\Delta^{t-1} = W^t - W^{t-1}$
 4:     $\beta_m^{t,0} = \beta^t$
 5:     **for** $k$ in $0, 1, ..., K-1$ **do**
 6:         Compute $g_m(W^{t,k})$ on $W^{t,k}$ and $\xi$,
 7:         Update local learning rate: $\beta_m^{t,k} = \beta_m^{t,k-1} + g_m(W^{t,k}) \cdot g_m(W^{t,k-1}) \cdot (1 + \varepsilon \cdot \frac{g_m(W^{t,k}) \cdot \Delta^{t-1}}{|g_m(W^{t,k}) \cdot \Delta^{t-1}|})$
 8:         Clip: $\beta_m^{t,k} = \min\{\max\{\beta_m^{t,k}, \frac{1}{\gamma_\beta}\}, \gamma_\beta\}$
 9:         Local model update: $W_m^{k+1} = W_m^k - \beta_m^{t,k} \cdot g_m(W^{t,k})$
10:     **end for**
11:     Send $\Delta_m^t = W_m^{t,K} - W^t$ to server.

**Server: FedHyper-G**

12:     Compute global model update: $\Delta^t = \Sigma P_m \Delta_m^t$
13:     Update global learning rate: $\alpha^t = \alpha^{t-1} + \Delta^t \cdot \Delta^{t-1}$
14:     Clip: $\alpha_t = \min\{\max\{\alpha_t, \frac{1}{\gamma_\alpha}\}, \gamma_\alpha\}$
15:     Update global model: $W^{t+1} = W^t - \alpha^t \cdot \Delta^t$
16: **end for**

---

We do not encourage other combinations because we do not observe an obvious performance improvement on them. We believe that FEDHYPER-G + FEDHYPER-CL can achieve the best performance in our framework if the trainer has a generous budget.

## C ALGORITHM OF FEDHYPER

We obtain the full FEDHYPER algorithm and show the cooperation of FEDHYPER-G and FEDHYPER-CL in Algorithm 1. Note that FEDHYPER-SL uses the same hypergradient as FEDHYPER-G so it is not applied in order to simplify the algorithm.

