# OpenReview forum: "FedHyper: A Universal and Robust Learning Rate Scheduler for Federated Learning with Hypergradient Descent"
_ICLR.cc/2024/Conference — ICLR 2024 poster_

### Official Review · Reviewer_xGw7 · 2023-10-20

**Soundness:** 2 fair
**Presentation:** 3 good
**Contribution:** 2 fair
**Rating:** 5
**Confidence:** 3

**Summary:**

This paper proposes FedHyper, which adopts Hypergradients to dynamically adjust learning rate for global learning rate (LR), server-side local LR, and client-side local LR. The authors demonstrate the effectiveness of FedHyper on various datasets and models, including image classification and language modeling tasks. FedHyper is shown to outperform other FL learning rate scheduling algorithms in terms of convergence rate and final accuracy on various datasets and models.

**Strengths:**

The paper is well-written and easy to understand, with clear explanations and well-designed experiments. FEDHYPER is a versatile algorithm that can seamlessly integrate with and augment the performance of existing optimization algorithms. This versatility makes it a valuable tool for researchers and practitioners working in the field of FL.

**Weaknesses:**

The proposed FedHyper highly relies on the existing method, i.e., hypergradient. It's unclear why the learning rate can be updated following Eq. (4-5). It would be better to give the intuitive explanation for this. FedHyper does not discuss the issue of training cost in FL.

**Questions:**

- Why are the baseline methods different for the three settings in Figure 3?
- Could you please provide a more detailed and intuitive explanation of hypergradients? I’m looking for a thorough understanding, and I’m willing to give a higher score for a comprehensive explanation.
- Is the learning rate fixed for baseline methods like FedAvg?
- Since FedHyper requires updating the learning rate using current and previous time gradients, does it incur significantly higher computational and resource costs compared to other baseline methods?

**Details Of Ethics Concerns:**

null

---

> ### Author Response · Authors · 2023-11-15
> **Thank you for your insightful feedback and valuable questions regarding our manuscript. We appreciate the opportunity to clarify and expand on the points you've raised.**
>
> We acknowledge the concern about FedHyper’s reliance on the hypergradient method and the need for a clearer explanation of the learning rate update process as per Equations (4-5). We have thoroughly explained how hypergradient works with simple examples. We respectfully urge the reviewer to consider these explanations in the final assessment. Below, we address each of your concerns and questions in detail:
>
> >W1: The proposed FedHyper highly relies on the existing method, i.e., hypergradient. It's unclear why the learning rate can be updated following Eq. (4-5). It would be better to give an intuitive explanation for this.
>
> Eq. (4-5) is the hypergradient descent process on the learning rate $\eta$ in centralized training. In the $t$th epoch of standard learning process of SGD algorithm, we first compute the model gradient $\nabla F(w^{(t)})$ by gradient descent. Then, we use the product of current learning rate $\eta^{(t)}$ and the model gradient to update the model parameters. Here in FedHyper, we update $\eta^{(t)}$ each epoch. The updated learning rate $\eta^{(t)}$ at $t$th epoch is computed as the sum of the previous learning rate $\eta^{(t-1)}$ and the product of the model gradients at epoch $t$ and $(t-1)$. This is mathematically represented as:
>
> $\eta^{(t)} = \eta^{(t-1)} + \nabla F(w^{(t)}) \cdot \nabla F(w^{(t-1)})$
>
> Since $\nabla F(w^{(t)})$ and $\nabla F(w^{(t-1)})$ are matrices of the same dimension because they are all gradients on the same model, their element-wise product results in a scalar. This scalar can be directly added to the original learning rate for updating the learning rate.
>
> To further clarify, we provide an illustrative example here to show how hypergradient works. We assume that there is a simple optimization problem with initial learning rate $\eta^{(0)} = 0.1$, the model parameters have size 1x2.
> We assume that the gradients on the model in the first three epochs are:
>
> $\nabla F(w^{(0)}) = [0.2, 0.1]$
>
> $\nabla F(w^{(1)}) = [0.1, 0.1]$
>
> $\nabla F(w^{(2)}) = [-0.1, -0.1]$
>
> Thus, we start to schedule the learning rate in 1st epoch, we use the model gradient in the 0th and 1st epochs to update $\eta^{0}$ to $\eta^{1}$:
>
> $\eta^{(1)} = \eta^{(0)} + \nabla F(w^{(0)}) \cdot \nabla F(w^{(1)})$
>
> $=0.1 + [0.2, 0.1] · [0.1, 0.1]$
>
> $= 0.13$
>
> $\eta$ increases from 0.1 to 0.13 because the model gradients in the first two epochs have similar directions-i.e., all the items are positive.
>
> Then the learning rate in 2nd epoch will be updated by:
>
> $\eta^{(2)} = \eta^{(1)} + \nabla F(w^{(1)}) \cdot \nabla F(w^{(2)})$
>
> $=0.13 + [0.1, 0.1] · [-0.1, -0.1]$
>
> $= 0.11$
>
> $\eta$ decreases from 0.13 to 0.11 because the 1st and 2nd epochs have opposite gradients.
>
> Specifically, in FL, we adapt this update rule to global and local learning rates, and use global model updates to replace the gradient in FedHyper-G and FedHyper-SL, use local gradients to replace the gradient in FedHyper-CL.
>
> >W2: FedHyper does not discuss the issue of training costs in FL.
>
> We appreciate the emphasis on the importance of training costs in FL. To address this, we have included a detailed analysis of the time cost of our three schedulers and baselines. We run the experiments on time cost of our three schedulers and baselines. The following results are the time cost (in seconds) of different global optimization algorithms when training CIFAR10 for 200 communication rounds:
>
> =======================================
>
> FedHyper-G $\quad$ FedAvg $\quad$ FedAdam $\quad$ FedExp
>
> ...................................................................................
>
>   $\quad$ 44060 $\quad$ $\quad$ 44031 $\quad$ $\quad$ 44045 $\quad$$\quad$ 45193
>
> =======================================
>
> We can see that the different global schedulers have similar running time largely due to the dominant time consumption of the local training process, while global updates and optimization constitute only a small fraction of the overall overhead. However, consider that FedHyper-G can reach convergence faster than baselines, we will have less cost than our baselines ultimately.
>
> Then, here is the time cost (in seconds) of different local learning rate scheduling algorithms in the same setting:
>
> =======================================
>
> FedHyper-CL  $\quad$    FedAvg(SGD)   $\quad$  FedAvg(Adam)
>
> ...................................................................................
>
> $\quad$ 45780$\quad$$\quad$ $\quad$ $\quad$ 44031 $\quad$ $\quad$ $\quad$ 32369
>
> =======================================
>
> Surprisingly, Adam requires much less training time than SGD and FedHyper-CL although it requires more computing overhead. It may be because CUDA or PyTorch has specifically designed optimization algorithms for Adam. However, we can find that FedHyper-CL only increase less than 5% computation cost compared with FedAvg(SGD) but gets convergence up to 3 times faster than SGD. Therefore, FedHyper's faster convergence offsets this slight increase in computation time.

---

> ### Author Response · Authors · 2023-11-15
>
> >Q1: Why are the baseline methods different for the three settings in Figure 3?
>
> This is because the baselines are not universal schedulers, some of them are only global schedulers-i.e., FedExp, FedAdam, and FedAdagrad, and some are only client local schedulers-i.e., SGD and Adam. Thus, to ensure fairness, we only compare the FedHyper-G with the global scheduler baseline, FedHyper-CL with client's local scheduler baselines… etc.
>
> >Q2: Could you please provide a more detailed and intuitive explanation of hypergradients? I’m looking for a thorough understanding, and I’m willing to give a higher score for a comprehensive explanation.
>
> We direct the reviewer to the improved explanation in W1, where we have elaborated on the concept and practical application of hypergradients, supplemented by a simple, illustrative example.
>
> >Q3: Is the learning rate fixed for baseline methods like FedAvg?
>
> FedAvg has fixed global and local learning rates. Decay-G, Decay-SL, and Decay-CL are the FedAvg with learning rate decay on global, server-side local, and client-side local learning rates respectively. FedExp calculates a global learning rate for each round, thus its global learning rate is not fixed. FedAdam and FedAdagrad directly manipulate gradient data, hence, we no longer manipulate the learning rate. Instead, we adopt the optimal initial learning rates summarized through experiments in [1] for our three datasets.
>
> >Q4: Since FedHyper requires updating the learning rate using current and previous time gradients, does it incur significantly higher computational and resource costs compared to other baseline methods?
>
>  While it's true that FedHyper introduces additional computational steps in each epoch or communication round compared to the baseline FedAvg method, the key advantage lies in its ability to significantly accelerate convergence. This faster convergence means that FedHyper can achieve the desired level of accuracy with fewer training rounds and epochs, ultimately leading to a net saving in computational resources.To quantify this, we consider the total number of training epochs required for the global model to reach a specific accuracy as an intuitive metric of computational cost for FedHyper and the baseline methods. Here is the number of communication rounds needed to get 56% accuracy on CIFAR10 dataset:
>
> ==================================================================================
>
>  FedHyper-G $\quad$ FedHyper-SL $\quad$ FedHyper-CL $\quad$ FedAvg $\quad$ FedAvg(Adam) $\quad$ FedAdagrad $\quad$ FedExp
>
> ..............................................................................................................................................................................
>
> $\quad$ 121 $\quad$ $\quad$ $\quad$ $\quad$ 128 $\quad$ $\quad$ $\quad$  $\quad$ 104 $\quad$ $\quad$$\quad$ 156$\quad$$\quad$$\quad$$\quad$155 $\quad$ $\quad$$\quad$ $\quad$ $\quad$181 $\quad$ $\quad$ $\quad$ 125
>
> ==================================================================================
>
> Then, here is the number of communication rounds needed to get 50% accuracy on the Shakespeare dataset:
>
> ==================================================================================
>
>  FedHyper-G $\quad$ FedHyper-SL $\quad$ FedHyper-CL $\quad$ FedAvg $\quad$ FedAvg(Adam) $\quad$ FedAdagrad $\quad$ FedExp
>
> ..............................................................................................................................................................................
>
>  $\quad$ 249 $\quad$ $\quad$ $\quad$ $\quad$ 177 $\quad$ $\quad$ $\quad$  $\quad$ 51 $\quad$ $\quad$
>  $\quad$ 292$\quad$$\quad$$\quad$ $\quad$48 $\quad$ $\quad$$\quad$ $\quad$ $\quad$ 101 $\quad$ $\quad$ $\quad$ 336
>
> ==================================================================================
>
> We can get the result that FedHyper gets convergence with less rounds than baselines. From W2, we can find that FedHyper adds require less than 5% computing time than FedAvg, thus it will cost less time to get convergence, which means less computational cost and time cost.
>
> Reference:
>
> [1]Sashank Reddi, Zachary Charles, Manzil Zaheer, Zachary Garrett, Keith Rush, Jakub Konecnˇ y,` Sanjiv Kumar, and H Brendan McMahan. Adaptive federated optimization. arXiv preprint arXiv:2003.00295, 2020

---

> > ### Author Response · Authors · 2023-11-20
> > **Looking forward to feedbacks for our response**
> >
> > Thank you once again for your valuable comments. As the discussion stage is coming to a close in 2 days, we kindly request your feedback on whether our response adequately addresses your concerns. We would greatly appreciate any additional feedback you may have.

---

> > > ### Author Response · Authors · 2023-11-21
> > > **Looking forward to feedbacks for our response**
> > >
> > > Thanks very much for your time and valuable comments. We understand you're busy. But as the window for responding and paper revision is closing, would you mind checking our response (a brief summary, and details) and confirming whether you have any further questions? We are very glad to provide answers and revisions to your further questions.
> > >
> > > Best regards and thanks,
> > >
> > >  Authors of #2230

---

> > ### Author Response · Authors · 2023-11-22
> > **Window for responding and draft updating is closing**
> >
> > Thanks very much for your time and valuable comments. We understand you're busy. But as the window for responding and paper revision is closing, would you mind checking our response and confirming whether you have any further questions? We are very glad to provide answers and revisions to your further questions.

---

### Official Review · Reviewer_EDET · 2023-10-30

**Soundness:** 4 excellent
**Presentation:** 4 excellent
**Contribution:** 4 excellent
**Rating:** 8
**Confidence:** 5

**Summary:**

This paper introduces FedHyper, a novel learning rate scheduler adept at managing both global and local learning rates. FedHyper distinguishes itself by accelerating convergence and automating learning rate adjustments. To validate FedHyper's efficacy, the authors perform thorough experiments on benchmark datasets, offering a comprehensive evaluation of its performance.

**Strengths:**

1) The paper addresses a significant issue in Federated Learning (FL) by emphasizing the pivotal role of learning rate scheduling. It successfully argues the necessity for meticulous attention in this area and proposes pragmatic solutions, thereby contributing tangibly to advancements in FL.
2) The adoption of the hypergradient method within this context is noteworthy. Its proven effectiveness in FL not only reinforces the method's utility but also broadens its appeal, suggesting it could be beneficial in a variety of other areas. This aspect of the paper stands out as particularly insightful.
3) The experimental framework of the paper is commendable for its solidity. By utilizing benchmark datasets, the research provides ample evidence to support the stated contributions of FedHyper. This rigorous approach to experimentation substantiates the claims made, enhancing the paper's credibility.

**Weaknesses:**

1) One concern is the lack of empirical evidence supporting the client-side scheduler's strategy of employing a global model update to limit the growth of the local learning rate. The paper would greatly benefit from an ablation study to confirm the validity of this approach, ensuring that the strategy is both necessary and effective.
2) The analysis seems incomplete when it comes to comparing FedHyper with FedAdam. The latter, considered a baseline, is conspicuously missing from the "performance of FedHyper" section after being included in the cooperation analysis. The reason for this omission is unclear, and it restricts a full understanding of how FedHyper stands against established methods.
3) The paper falls short in explaining the underlying reasons behind FedHyper's superior ability to fine-tune learning rates compared to its contemporaries, such as FedExp. The missing intuitive rationale or clear justification leaves the reader questioning why FedHyper is ostensibly more efficient. Addressing this would make the advantages of FedHyper more transparent and convincing.

**Questions:**

Please refer to Weakness.

---

> ### Author Response · Authors · 2023-11-17
>
> >W1: One concern is the lack of empirical evidence supporting the client-side scheduler's strategy of employing a global model update to limit the growth of the local learning rate. The paper would greatly benefit from an ablation study to confirm the validity of this approach, ensuring that the strategy is both necessary and effective.
>
> To show that the added item with global model update can improve the performance of FedHyper-CL, we add an ablation study on the item in different $\alpha$ in non-iid data distribution. Here are the results:
>
> =====================================
>
> $\quad$ $\quad$ $\quad$ $\quad$ $\quad$ $\quad$ 0.2 $\quad$ $\quad$ 0.5 $\quad$ $\quad$ 0.75
>
> ...............................................................................
>
> With item  $\quad$$\quad$  55.10% $\quad$58.42% $\quad$57.18%
>
> Without item$\quad$53.62% $\quad$57.79% $\quad$57.12%
>
> =====================================
>
> We observed that the added term primarily improves model training accuracy when the non-iid degree is high-i.e., low $\alpha$ value. This aligns with our intuition for incorporating this term - to enhance the effectiveness of heterogeneous client scenarios.
>
> >W2: The analysis seems incomplete when it comes to comparing FedHyper with FedAdam. The latter, considered a baseline, is conspicuously missing from the "performance of FedHyper" section after being included in the cooperation analysis. The reason for this omission is unclear, and it restricts a full understanding of how FedHyper stands against established methods.
>
> We thank the reviewer for pointing out an oversight in our experiments. According to FedExp, the FedAdam should be compared with algorithms incorporating server momentum methods. Thus, here we provide the comparison between FedHyper-G+Server Momentum and FedAdam and the result is as follows:
>
> CIFAR10 200epochs:
>
> ========================
>
> FedHyper-GM $\quad$ FedAdam
>
> ..................................................
>
>  $\quad$ 58.66  $\quad$$\quad$$\quad$$\quad$ 47.61
>
> ========================
>
> Shakespeare 600epochs:
>
> ========================
>
> FedHyper-GM $\quad$ FedAdam
>
> ..................................................
>
>  $\quad$ 53.44 $\quad$ $\quad$$\quad$$\quad$ 50.98
>
> ========================
>
> FedHyper-GM outperforms FedAdam in accuracy.
>
> >W3: The paper falls short in explaining the underlying reasons behind FedHyper's superior ability to fine-tune learning rates compared to its contemporaries, such as FedExp. The missing intuitive rationale or clear justification leaves the reader questioning why FedHyper is ostensibly more efficient. Addressing this would make the advantages of FedHyper more transparent and convincing.
>
> We thank the reviewer for suggestions on adding intuitive explanations. Here we provide an intuitive explanation on why FedHyper can outperform FedExp in some situations. FedExp tends to use a larger global learning rate when the local model updates have higher variance. However, due to the fact that the local model update exhibits a smaller variance in the early stages of training and a larger variance as it approaches convergence, FedExp thus may has a higher global learning rate in the late stage than in the beginning of the training process, which can be observed in Figure 6 in our appendix. This contradict with our intuition that the learning rate should be large at the beginning and be low in the late stage. Thus, FedExp cannot work consistently stable across different tasks. It may only work well on tasks that need high global learning rates.

---

### Official Review · Reviewer_aiMM · 2023-10-31

**Soundness:** 3 good
**Presentation:** 3 good
**Contribution:** 3 good
**Rating:** 8
**Confidence:** 4

**Summary:**

This paper aims to introduce a structured approach to learning rate scheduling within the realm of Federated Learning, a critical step that enhances efficiency, particularly in the initialization phase. The authors present FedHyper, a framework that leverages hypergradient methods to adjust learning rates based on the inner product of gradients. FedHyper encompasses three distinct schedulers for comprehensive application: a global scheduler for the server-side learning rate, a server-side local scheduler, and a client-side local scheduler. Through rigorous testing on three datasets, FedHyper demonstrates superior performance compared to state-of-the-art baselines.

**Strengths:**

- By targeting a pivotal issue in Federated Learning, the study positions itself within a crucial niche. The focus on optimizing learning rate scheduling addresses a substantive bottleneck in the field, underlining the paper's relevance.
- The hypergradient method is interesting and effective, and the proposed method could be applied to other hyperparameters as well. Simplicity and practicality are the hallmarks of the proposed scheduler, making it an attractive tool for real-world application. Its ease of use could significantly benefit practitioners in the field.
- The convergence of FedHyper is theoretically proofed, making this work more sound.
- The results from extensive experiments demonstrate the significant performance improvement in both convergence rate and final accuracy.
- The paper excels in clarity and accessibility, effectively illustrating the mechanics of hypergradient descent in learning rate scheduling through well-conceptualized figures (e.g., Figures 1 and 2).

**Weaknesses:**

- The paper's primary methodology involves utilizing the inner product of gradients to modify the learning rate, a technique that potentially incurs additional computational expenses compared to the more basic FedAvg. The study falls short by not evaluating or discussing these potential overheads, leaving the reader uncertain about the practical trade-offs.
- In FedHyper-CL, the author’s statement “directly applying Eq. (19) to local learning rates can lead to an imbalance in learning rates across clients” is not well supported by analysis or reference. So, the motivation of adding an item in FedHyper-CL is not clear.
- The paper does not provide the direct comparison with FedAdam.

**Questions:**

See the weaknesses.

---

> ### Author Response · Authors · 2023-11-17
>
> The authors would like to thank the reviewers for their time and especially for all of the thoughtful questions and constructive suggestions. We would like to share our responses below.
>
> >W1: The paper's primary methodology involves utilizing the inner product of gradients to modify the learning rate, a technique that potentially incurs additional computational expenses compared to the more basic FedAvg. The study falls short by not evaluating or discussing these potential overheads, leaving the reader uncertain about the practical trade-offs.
>
> We appreciate the emphasis on the importance of training costs in FL. To address this, we have included a detailed analysis of the time cost of our three schedulers and baselines. We ran the experiments on time cost of our three schedulers and baselines. The following results are the time cost (in seconds) of different global optimization algorithms when training CIFAR10 for 200 communication rounds:
>
> =======================================
>
> FedHyper-G $\quad$ FedAvg $\quad$ FedAdam $\quad$ FedExp
>
> ...................................................................................
>
>   $\quad$ 44060 $\quad$ $\quad$ 44031 $\quad$ $\quad$ 44045 $\quad$$\quad$ 45193
>
> =======================================
>
> Different global schedulers have similar running times largely due to the dominant time consumption of the local training process, while global updates and optimization constitute only a small fraction of the overall overhead. However, our FedHyper-G can reach convergence faster than baselines, so we will have less cost than our baselines ultimately.
>
> Then, here is the time cost (in seconds) of different local learning rate scheduling algorithms in the same setting:
>
> =======================================
>
> FedHyper-CL  $\quad$    FedAvg(SGD)   $\quad$  FedAvg(Adam)
>
> ...................................................................................
>
> $\quad$ 45780$\quad$$\quad$ $\quad$ $\quad$ 44031 $\quad$ $\quad$ $\quad$ 32369
>
> =======================================
>
> Adam requires much less training time than SGD and FedHyper-CL although it requires more computing overhead may be because CUDA or PyTorch has specifically designed optimization algorithms for Adam. FedHyper-CL only increases less than 5% computation cost compared with FedAvg(SGD) but gets convergence up to 3 times faster than SGD. Therefore, FedHyper's faster convergence offsets this slight increase in computation time.
>
> >W2: In FedHyper-CL, the author’s statement “directly applying Eq. (19) to local learning rates can lead to an imbalance in learning rates across clients” is not well supported by analysis or reference. So, the motivation for adding an item in FedHyper-CL is not clear.
>
> In heterogeneous federated learning, different clients have datasets in different distributions. Some of the local datasets are easy to train and some are hard. If we do not add the item, the local learning rate will quickly increase in the clients with easy datasets and decrease in the clients with hard datasets. This is the reason of adding the item. The impact of heterogeneous clients can be found in [1] and [2].
>
> >W3: The paper does not provide a direct comparison with FedAdam.
>
> We thank the reviewer for pointing out an oversight in our experiments. According to FedExp, the FedAdam should be compared with algorithms incorporating server momentum methods. Thus, here we provide the comparison between FedHyper-G+Server Momentum and FedAdam and the result is as follows:
>
> CIFAR10 200epochs:
>
> ========================
>
> FedHyper-GM $\quad$ FedAdam
>
> ..................................................
>
>  $\quad$ 58.66  $\quad$$\quad$$\quad$$\quad$ 47.61
>
> ========================
>
> Shakespeare 600epochs:
>
> ========================
>
> FedHyper-GM $\quad$ FedAdam
>
> ..................................................
>
>  $\quad$ 53.44 $\quad$ $\quad$$\quad$$\quad$ 50.98
>
> ========================
>
> FedHyper-GM outperforms FedAdam in accuracy.
>
> References:
>
> [1]Xiang Li, Kaixuan Huang, Wenhao Yang, Shusen Wang, and Zhihua Zhang. On the convergence of fedavg on non-iid data. arXiv preprint arXiv:1907.02189, 2019.
>
> [2]Jianyu Wang, Qinghua Liu, Hao Liang, Gauri Joshi, and H Vincent Poor. Tackling the objective inconsistency problem in heterogeneous federated optimization. Advances in neural information processing systems, 33:7611–7623, 2020.

---

### Official Review · Reviewer_MWyj · 2023-11-02

**Soundness:** 4 excellent
**Presentation:** 4 excellent
**Contribution:** 4 excellent
**Rating:** 8
**Confidence:** 4

**Summary:**

This work extends the traditional hypergradient learning rate scheduler to federated learning scenarios. Specifically, the authors propose a novel theoretical framework that jointly considers the global and local learning rates with respect to global and local updates. The authors have conducted sufficient experiments to validate the performance.

**Strengths:**

1. The paper is well written. The authors thoroughly reviewed the previously related work, hypergradient, and discussed the limitations of the work in detail. Then the authors naturally extended the method into federated learning scenarios.

2. The proposed method is intuitive, and the theoretical guarantee is solid.

3. The authors have conducted sufficient experiments to validate the proposed methods and discussed the suitability of several variants under different hardware constraints.

**Weaknesses:**

The proposed method is only evaluated on simple datasets and tasks, such as CIFAR-10 and FMNIST, and tested with small models. The hyperparameter choices for these simple scenarios would be relatively straightforward, particularly in the case of standard centralized training. It would be beneficial if the authors could test the performance improvements on more challenging cases, such as unbalanced or large datasets, or in fine-tuning settings.

**Questions:**

FedHyper-SL and FedHyper-G appear to update different sets of hyperparameters using the same gradients. It would be beneficial if the authors could provide a more detailed comparison between these two variants, considering that FedHyper-G seems to be a special case of FedHyper-SL.

---

> ### Author Response · Authors · 2023-11-21
> **Summary: We appreciate the reviewer's thoughtful feedback on our work. We address all the concerns in detail below.**
>
> >W1: The proposed method is only evaluated on simple datasets and tasks, such as CIFAR-10 and FMNIST, and tested with small models. The hyperparameter choices for these simple scenarios would be relatively straightforward, particularly in the case of standard centralized training. It would be beneficial if the authors could test the performance improvements on more challenging cases, such as unbalanced or large datasets, or in fine-tuning settings.
>
> To test the performance improvements of FedHyper on more challenging cases, we adopt FedHyper-G on federated fine-tuning large language models. We use a Llama-7b base model to fine-tune the Databricks-dolly-15k dataset and test it on the MMLU test set. We use LoRA to fine-tune the local models for 5 communication rounds and 2 local epochs each round. We get the following results:
>
> ======================
>
> $\quad$FedAvg $\quad$ FedHyper-G
>
> ..............................................
>
>  $\quad$ 32.74$\quad$$\quad$$\quad$ 33.18
>
> ======================
>
> This result shows that FedHyper can also bring benefits to the fine-tuning of large language models. Considering that the local optimizer of fine-tuning llms is a more complicit problem, we leave adopting FedHyper-CL to llm fine-tuning a future topic.
>
> >Q1: FedHyper-SL and FedHyper-G appear to update different sets of hyperparameters using the same gradients. It would be beneficial if the authors could provide a more detailed comparison between these two variants, considering that FedHyper-G seems to be a special case of FedHyper-SL.
>
> We highly appreciate the reviewer's perspective that FedHyper-G is a special case of FedHyper-SL. According to our observation, FedHyper-SL outperforms FedHyper-G in most of the task because the local learning rate takes effect in every local epoch while the global learning rate only works once per round. This is the comparison of FedHyper-G and FedHyper-SL and FedHyper-G+FedHyper-SL in our three dataset:
>
> FMNIST:
>
> =====================================
>
> $\quad$FedHyper-G FedHyper-SL FedHyper-G+SL
>
> ..............................................................................
>
>  $\quad$ $\quad$ 97.04  $\quad$ $\quad$ 97.01 $\quad$ $\quad$ $\quad$ 96.90
>
> =====================================
>
> CIFAR10:
>
> =====================================
>
> $\quad$FedHyper-G FedHyper-SL FedHyper-G+SL
>
> ..............................................................................
>
>  $\quad$ $\quad$ 57.44 $\quad$ $\quad$ 58.35 $\quad$ $\quad$ $\quad$ 58.37
>
> =====================================
>
> Shakespeare:
>
> =====================================
>
> $\quad$FedHyper-G FedHyper-SL FedHyper-G+SL
>
> ..............................................................................
>
>  $\quad$ $\quad$ 51.42  $\quad$ $\quad$ 51.89 $\quad$ $\quad$ $\quad$ 52.56
>
> =====================================
>
> We can find that the performance of FedHyper-G and FedHyper-SL vary for different tasks. In FMNIST and CIFAR10, the performance of FedHyper-G and FedHyper-SL are similar to each other, there is also no performance improvement in FedHyper-G+SL. In the Shakespeare dataset, which is more sensitive to local learning rate, FedHyper-SL and FedHyper-G+SL outperform FedHyper-G. This indicates that for most of the simple tasks, FedHyper-G is enough and has the least computation and communication overhead. For local learning rate sensitive tasks, we recommend using FedHyper-SL.

---

> > ### Author Response · Authors · 2023-11-21
> > **Looking forward to feedbacks for our response**
> >
> > Thank you once again for your valuable comments. As the discussion stage is coming to the end, we kindly request your feedback on whether our response adequately addresses your concerns. We would greatly appreciate any additional feedback you may have.

---

> > > ### Comment · Reviewer_MWyj · 2023-11-21
> > >
> > > Thank you for your detailed answers! Most of my concerns were addressed, I increase my score to 8.

---

> > > > ### Author Response · Authors · 2023-11-21
> > > > **Thanks**
> > > >
> > > > Thank you very much for your feedback and for taking the time to re-evaluate our submission. We are pleased to hear that our responses have addressed most of your concerns, and we greatly appreciate the improved score of 8. Once again, thank you for your constructive feedback and for the positive reassessment of our work.

---

### Meta-Review · Area_Chair_Qba2 · 2023-12-06

**Metareview:**

Overall, the reviewers recognize this work's strong potential in advancing federated learning by tackling the pivotal challenge of adaptively scheduling learning rates for both global and local updates. All reviewers praised the method's simplicity, effectiveness, theoretical grounding, and practical appeal. Extensive experiments demonstrate clear convergence and accuracy improvements over existing techniques.The authors have successfully addressed most concerns during rebuttal, leading to raised scores.

I believe the consistently high appraisal across other dimensions, combined with the largely constructive nature of criticisms focused on supplemental analysis and details, supports acceptance. Please address the mentioned additional experiments and explanations during revisions.

**Justification For Why Not Higher Score:**

While generally positive, concerns existed regarding additional computational overhead, motivation for certain design choices, and comparisons to related methods like FedAdam. Two reviewers also want to see evaluations on more complex tasks and models. Additionally, one reviewer rated the core technical novelty as incremental upon prior hypergradient approaches.

**Justification For Why Not Lower Score:**

The work clearly defines an intuitive adaptation to make learning rate scheduling - a stubborn pain point in FL - tractable through principled optimization. The general sentiment of reviews is very positive.

---

### Decision · Program_Chairs · 2024-01-16

Accept (poster)